# Ubiquitous organic molecule-based free-standing nanowires with ultra-high aspect ratios

Koshi Kamiya[1], Kazuto Kayama[1], Masaki Nobuoka[1], Shugo Sakaguchi[1], Tsuneaki Sakurai [1✉], Minori Kawata[1], Yusuke Tsutsui[1], Masayuki Suda [1], Akira Idesaki[2], Hiroshi Koshikawa[2], Masaki Sugimoto[2], G. B. V. S. Lakshmi[3], D. K. Avasthi[4] & Shu Seki [1✉]

The critical dimension of semiconductor devices is approaching the single-nm regime, and a variety of practical devices of this scale are targeted for production. Planar structures of nano-devices are still the center of fabrication techniques, which limit further integration of devices into a chip. Extension into 3D space is a promising strategy for future; however, the surface interaction in 3D nanospace make it hard to integrate nanostructures with ultrahigh aspect ratios. Here we report a unique technique using high-energy charged particles to produce free-standing 1D organic nanostructures with high aspect ratios over 100 and controlled number density. Along the straight trajectory of particles penetrating the films of various sublimable organic molecules, 1D nanowires were formed with approximately 10~15 nm thickness and controlled length. An all-dry process was developed to isolate the nanowires, and planar or coaxial heterojunction structures were built into the nanowires. Electrical and structural functions of the developed standing nanowire arrays were investigated, demonstrating the potential of the present ultrathin organic nanowire systems.

[1] Department of Molecular Engineering, Graduate School of Engineering, Kyoto University, Kyoto, Japan. [2] Takasaki Advanced Radiation Research Institute, National Institutes for Quantum and Radiological Science and Technology, Takasaki, Gunma, Japan. [3] Special Center for Nanoscience, Jawaharlal Nehru University, New Delhi, India. [4] Department of Physics, School of Engineering, University of Petroleum and Energy Studies, Dehradun, India. ✉email: sakurai-t@moleng.kyoto-u.ac.jp; seki@moleng.kyoto-u.ac.jp

Uniform low-dimensional nanostructures—nanoparticles, nanosheets, and nanowires—have gained attention as an important class of materials contributing to the progress in the research fields of materials science and nanotechnology[1–5]. Nanowires offer extra-large specific surface areas useful for sensing platforms[6], anisotropic emissions[7,8], and the ability of one-dimensional (1D) transport of charges, energy, heat, and so on[9–11]. To realize anisotropic transport along nanowires at a macroscopic scale, the nanowires need to be aligned while maximizing their in-plane density—if they lack unidirectional orientation, the effects of anisotropy are lost for the entire system. In this context, a vertical array of uniform ultrathin nanowires is an ideal candidate for such platforms, providing both extra-large surface area and highly aligned unidirectional 1D pathways. Various fabrication methods of inorganic nanowire arrays have been established so far, as represented by the top-down lithography[12,13] and bottom-up epitaxial growth approaches[14,15]. For example, uniform standing nanowires based on silicon were fabricated with diameters of ~10–100 nm, and their semiconducting properties were demonstrated by integration into a field-effect transistor device[16]. Another noteworthy example is a vertically aligned carbon nanotube (VA-CNT) array grown by the chemical vapor deposition, with adequate carbon source and a catalytic seed-deposited substrate[17,18]. Unidirectional alignment of VA-CNTs, coupled with their unique optical and mechanical characteristics, enables their application, such as black absorbers and adhesive materials[19,20]. In contrast to these hard matter-based examples, standing organic nanowires have as yet been unexplored. Although bottom-up crystalline growth has been successful in fabricating VA-nanowires from several organic molecules, they have the drawback of low structural purity—being contaminated with different nanostructures grown during the growth process[21,22]. They have two additional problems in that the aspect ratio of the VA-nanowires is hard to tune by changing the growth conditions and their diameters are limited to several hundreds of nm (submicron scale). Standing nanowires have also been achieved from polymer materials through lithography, but the reported highest aspect ratio was only as high as ~10[23]. A template method using an anodic aluminum oxide (AAO) with cylindrical nanopores was reported, which enables the growth of thin nanowires from organic small molecules with minimum diameter of ~50 nm[24]. However, these nanowires fell over upon removal of the AAO template by a wet process, eventually resulting in the in-plane orientation of the nanowires. Therefore, there has been a strong desire for widely applicable techniques to develop standing NWs with ultrahigh aspect ratios from a variety of organic materials, which will expand their application in optics, imaging, electronics, bioelectronic, and mechanical fields.

We have developed a technique of nanowire fabrication, referred to as single particle-triggered linear polymerization (STLiP), fabricating uniform nanowires while completely controlling their length, diameter, and number density (Fig. 1). In this method, a single charged particle, regarded as the finest stream of energy, triggers a chemical reaction by depositing its kinetic energy within a nanometer-scale small spatial area. When a high-energy charged particle passes through a condensed organic layer, the energy of the particle is transferred to a limited cylindrical region of the layer, defined as the ion track region[25,26]. Within the ion track, nonuniform energy transfer occurs and reaction intermediates, such as neutral radicals, are germinated[27,28]. The intermediates induce cross-linking/polymerization reactions to form 1D gels, resulting in the formation of organic nanowires. The striking advantage of the STLiP method, compared with the other conventional ones, is its applicability for a wide variety of materials as targets. In principle,

heavy charged particles, such as Xe, follow a straight trajectory in condensed phases of organic substances because of the negligible momentum/energy transfer from the kinetic energy of incident particles[29]. This negligible change in the momentum of the particle inspired us to address the production of standing nanowires on a substrate by introducing the particles orthogonally (Fig. 1). Immersion of the irradiated film to dissolve the unirradiated part of the organic substances has been the usual choice for the isolation of latent nanowires on the substrate: the wet development process as used typically in conventional nanofabrication techniques. It causes, however, knocking-down of the resulting nanowires on the substrate due to the irreversible adhesion of nanowires on the substrate via the surface tension force of the solvent[30]. Although supercritical fluid is an alternative for organic solvents in the wet process[31], the isolation of uniform nanowires with high aspect ratio without the collapse of the pattern is still unexplored. Here, we achieve the isolation of standing organic nanowires with ultrahigh aspect ratios by a dry process: the sublimation of unreacted organic molecules. The direct phase transition from solid to gas phases for the nonirradiated area in the films results in the successful isolation of standing nanowires based on polymerized organic materials, with dense distribution and ultrahigh aspect ratio. Further functionalization of nanowires—the design of heterointerfaces[32,33]—was demonstrated in this work. Two basic types of heterointerface nanostructures have been proposed thus far, namely, coaxial design[32–35] and two adjoining segments[32,33,36–38]. The reported studies are categorized into AAO templated[35,37,38] and physical vapor transport[34,36] methods. From another perspective, they are classified into inorganic–inorganic[32], organic–inorganic[35,37], and organic–organic[34,36,38] nanowire systems.

Herein, electropolymerization around standing nanowires and block-*co*-nanowire approaches using bilayer films are explored to realize the respective heterointerfaces in our ultrathin organic nanowire systems. Namely, the tunable extension of wires in both the longitudinal and radial directions is established using a variety of organic substances, illustrating the full potential of our newly developed technology.

## Results

**Fabrication of standing organic nanowires**. The nanowires were fabricated along particle trajectories as illustrated in Fig. 1, coexisting with the major fraction of unreacted organic molecules far from the trajectories. Local polymerization reactions in the latent nanowires causes a significant decrease of solubility in organic solvents due to the increase in molecular weight. The latent nanowires could be easily isolated by a wet process with selective dissolution of the unreacted molecules. However, collapsing and aggregation of nanowires has generally been unavoidable, owing to solvent effects. The strong interactions between the nanowire and substrate surfaces irreversibly fixed the nanowires on the substrate[39], leading to their random distribution with a global worm-like chain configuration, which showed the flexibility of nanowires in the presence of the solvents, and the configuration could be interpreted based on the scaled macromolecular chain configuration in dilute solutions[40]. Macromolecular systems are also superior materials to promote the creation of latent nanowires in condensed phases via gelation, where insolubility of the macromolecules can be achieved with an average of one crosslink point per polymer molecule[41]. In contrast, it is presumed to be difficult to obtain nanowires, in which perfect immobilization of molecules occurs along the corresponding particle trajectory in the STLiP protocol. Recently, our studies revealed that highly efficient polymerization/cross-linking reactions, such as radical chain reactions in condensed phases, are available to afford nanowire

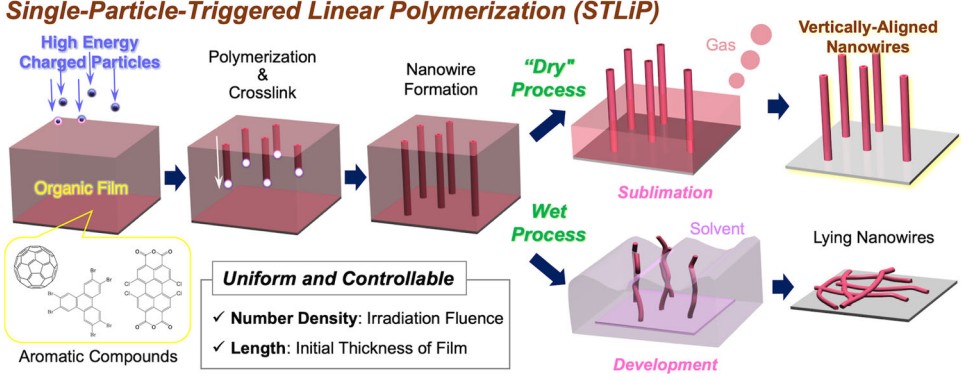

**Fig. 1 Nanowire fabrication process.** Schematic illustration of fabrication and isolation of knocked-down(lying)/standing(vertically-aligned) organic nanowires via polymerization reactions in ion tracks triggered by irradiation with high-energy charged particles (STLiP method).

formation via the STLiP protocol from a wide choice of small molecular materials[42,43].

In contrast to macromolecular systems, the higher vapor pressure of small organic molecules often provides them with sublimation ability, which provides us an overwhelming advantage in postirradiation processing. Polymerization reactions in the latent nanowires eliminate the sublimation ability of the initial molecular materials, which prompted us to selectively remove the unreacted molecules by simple heating in a vacuum, and thus isolate the nanowires on the substrates (Fig. 1). This sublimation ability is, in general, often utilized for fabrication of organic thin films, such as by vapor deposition. In contrast, here the sublimation is utilized for the development in postirradiation processes to remove excess unreacted material; this is an innovative point of the present work. In this all-dry process, we no longer consider the interaction between solvents and nano-patterns (nanowires), and/or their collapse during the process. No surface tension operates on the VA-nanowires, enabling us to obtain standing nanowires with ultrahigh aspect ratio free from pattern collapse.

As a target small molecular material, the Buckminster fullerene $C_{60}$ molecule was selected; $C_{60}$ is a fully isotropic organic molecule that has equivalent unsaturated carbon bonds[44]. $C_{60}$ has been reported to be sublimable, and its film is easily fabricated by vapor deposition[45], an important requisite in all-dry postirradiation STLiP protocols. Besides, $C_{60}$ is a representative molecule to facilitate chain polymerization reactions, initiated upon exposure to photo- and ionizing radiation[46,47]. The polymerization reaction has been well confirmed and characterized by conductive atomic force microscopy (AFM) and Raman scattering measurements[48,49]. We have also previously reported the formation of randomly spread $C_{60}$ nanowires in 2D as a polymeric material obtained after a chain polymerization reaction, isolated by a wet process using organic solvent[50]. Hence, in this paper, we attempted to isolate $C_{60}$ nanowires by an all-dry process. $C_{60}$ thin films were prepared by vapor deposition at 250–2000 nm thickness on Si substrates. Figure 2 shows the morphology of isolated nanowires observed by scanning electron microscopy (SEM), illustrating the striking contrasts in the images taken after the all-dry and wet isolation processes. After the isolation of the nanowires by the dry process, standing nanowires were clearly confirmed without any collapse, with dense distributions as high as ~$10^{11}$ cm$^{-2}$ (Fig. 2a–c). In contrast, significant collapse of knocked-down nanowires was observed on the substrate after isolation by a wet process (Fig. 2d), suggesting the presence of strong inter-nanowire and nanowire–substrate interactions promoted by solvents. Observation of the morphology by AFM also supported the supposition that the nanowires

unavoidably aggregate during the wet process, as shown in the white area in Supplementary Fig. 1a, losing their verticality. On the other hand, the total length of the nanowires strictly reflected the initial thickness of the vapor-deposited $C_{60}$ film in the dry process, verifying the production of standing nanowires with uniform and controllable length over a defined area (Supplementary Fig. 1c, d). It is noteworthy that the vertical alignment of nanowires was maintained clearly over their full length of 2500 nm (Fig. 2c), which demonstrated the production of organic molecular-based nanostructures with ultrahigh aspect ratios over 300. Particles of 350 MeV $^{129}$Xe$^{26+}$ maintain high linearity in $C_{60}$ solid films ($\rho = 1.72$ g cm$^{-3}$) as thick as 10 μm in trajectories simulated by a Monte Carlo method (SRIM 2010, Supplementary Fig. 2). This linearity guarantees that their energy transfer to the target is negligible in comparison with their initial kinetic energy in this regime. The transferred energy is estimated to reach up to 10% of the initial energy, suggesting that vertical nanowires as long as 20 μm can theoretically be achieved.

**Molecules for standing nanowire fabrication.** The isolation of nanowires by the dry process can minimize the effects of solvents. However, another important factor should be considered. The vertical alignment of the nanowires deteriorated when the irradiation fluence was reduced to $5.0 \times 10^{10}$ and $1.0 \times 10^{10}$ cm$^{-2}$ (Supplementary Fig. 3). Inter-wire interactions in the nanowire plexus maintain the assembled structures on the substrate; at reduced in-plane number densities of nanowires, the interactions are not sufficient to keep the nanowires standing. Then, to confirm the effect of solvent independently, we treated the once-isolated standing nanowire plexus with an organic solvent (Supplementary Fig. 4). As can be seen in Supplementary Fig. 4a–c, the vertical alignment of the nanowire gradually disappeared with immersion time in the solvent, showing local aggregation with random domain sizes. The issue has been recognized in nano-lithography processes as pattern collapse, particularly for structures with high aspect ratio[51]. The observation also suggests that the terminus of the nanowires is tightly connected onto the substrate surfaces. The inter-nanowire interactions were much enhanced with increasing length, as clearly seen in Supplementary Fig. 4e, where the bright tone of the images represents the 2D-like nanowire assemblies. These detailed experimental results imply the great advantage of the dry process with sufficient in-plane number density (=irradiation fluence).

$C_{60}$ is a representative carbon allotrope, and its chemical modification/functionalization has been frequently reported. Here, we address whether the Buckminster fullerene framework is necessary to create self-standing nanowires; this is the key question regarding the versatility of the present STLiP protocol.

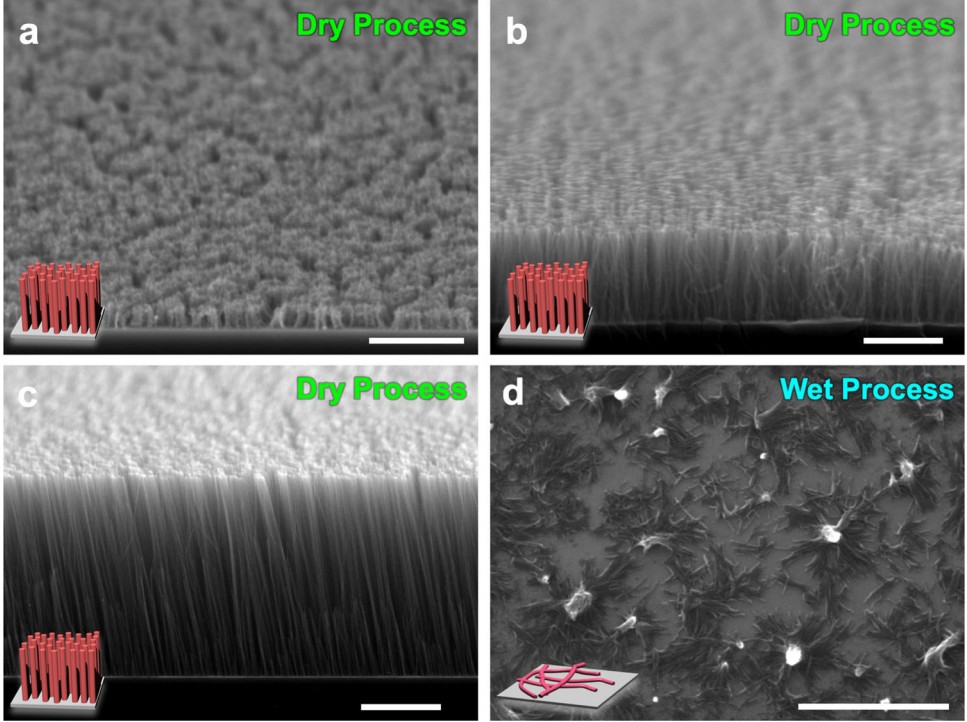

**Fig. 2 SEM images of standing or knocked-down nanowires of C$_{60}$.** SEM images of standing nanowires fabricated via charged particle irradiation of **a**, **d** 250 nm, **b** 1100 nm, and **c** 2500 nm thickness C$_{60}$ films, and isolated by **a–c** dry process: sublimation and **d** wet process: 1,2-dichlorobenzene. Scale bars represent 1 μm. Irradiation conditions: **a**, **d** 490 MeV $^{192}$Os$^{30+}$ at 1.0 × 10$^{11}$ cm$^{-2}$; **b**, **c** 350 MeV $^{129}$Xe$^{26+}$ at 1.0 × 10$^{11}$ cm$^{-2}$.

We attempted the formation of standing nanowires from a well-known modified fullerene derivative PC$_{61}$BM ([6,6]-phenyl-C$_{61}$-butyric acid methyl ester) and a larger fullerene C$_{70}$ (Supplementary Fig. 5). From PC$_{61}$BM, clear nanowires were obtained in both dry (Supplementary Fig. 5a–d) and wet (Supplementary Fig. 5e–h) processes. The melting point of PC$_{61}$BM, reported as ~280 °C (ref. [52]), is significantly lower than the sublimation temperature of 300 °C in the applied vacuum conditions, resulting in melting of the unreacted PC$_{61}$BM molecules to cause severe surface tension and eventually deform the self-standing nanowires. In sharp contrast, self-standing C$_{70}$ nanowires were developed clearly by the all-dry process, in spite of their reduced verticality compared to that of C$_{60}$. One possible explanation for this difference is the lower reactivity of C$_{70}$ upon heavy ion irradiation. In fact, comparing the cross-sectional radii of nanowires, the nanowires from C$_{70}$ showed a smaller cross-sectional radius than those from C$_{60}$, which suggests a higher polymerization efficiency for C$_{60}$.

Here, the reaction efficiency due to the energy transferred by particle irradiation is introduced, as the explanation for the change in cross-sectional radius. The energy transfer efficiency of radiation is often expressed in terms of linear energy transfer (LET), which has been proposed based on the equation for the spatial distribution of the energy density imparted to a material in an ion track, as a function of the distance from the track center $r$[53]:

$$\rho(r) = \frac{LET}{2}\left[2\pi r^2 \ln\left(\frac{e^{1/2}r_p}{r_c}\right)\right]^{-1},\qquad(1)$$

where $e$ is an exponential factor, and $r_c$ and $r_p$ are the radii of the core and penumbra areas of a particle track. Then, a simple assumption is applied, in which polymer cross-linking proceeds in the boundary region giving the nanowires (cross-sectional radius of the nanowires: $r_{cc}$). Generally, for gel formation in a

polymer system, it is necessary to introduce one crosslink per polymer molecule. By introducing $G(x)$, the reaction efficiency per 100 eV of irradiated particle energy, the required density of energy ($\rho_{cr}$) is given by,

$$\rho_{cr} = \frac{100dA}{G(x)mN},\qquad(2)$$

where $A$, $d$, $m$, and $N$ are Avogadro's number, the gravitational density of the solid polymer, mass of a monomer unit, and degree of polymerization, respectively. Adapting Eqs. (1) and (2) for low-molecular-weight materials, $G(x)$ is expressed by Eq. (3) below,

$$G(x) = \frac{200\pi dA r_{cc}^2}{LET \cdot M}\left[1 + 2\ln\left(\frac{r_p}{r_c}\right)\right],\qquad(3)$$

where $M$ is the molecular weight. Substituting the parameters of C$_{60}$ and C$_{70}$ into this equation, the $G$ values of C$_{60}$ and C$_{70}$ are 35.1 and 15.3, respectively, indicating that the reaction efficiency of C$_{60}$ is ~2.3 times higher than that of C$_{70}$. Dimerization reactions were reported both in C$_{60}$ and C$_{70}$ upon photo-exposure and irradiation to ionizing radiation[54–56]. The estimated ~3-fold higher efficiency in solid C$_{60}$ than in C$_{70}$ (ref. [57]) can well explain our experimental results for high-energy ion irradiation.

The breadth of the feasibility of the STLiP technique can be demonstrated by fabricating standing nanowires from a wide variety of organic molecules besides C$_{60}$ or C$_{70}$. We focused on conjugated small molecules as targets of STLiP protocols to promote efficient chain polymerization reactions. As demonstrated in Fig. 3, a variety of aromatic molecules suitable for the dry process with the ability to be sublimed under vacuum were selected as the targets, and clearly gave standing nanowires from the corresponding molecules. It is noteworthy that the nanowires could be produced from the simple sublimable polyaromatic hydrocarbon 9,10-bis(phenylethynyl)anthracene (BPEA), where the STLiP protocol could induce efficient chain polymerization/cross-linking reactions mediated by the reaction at C–C triple

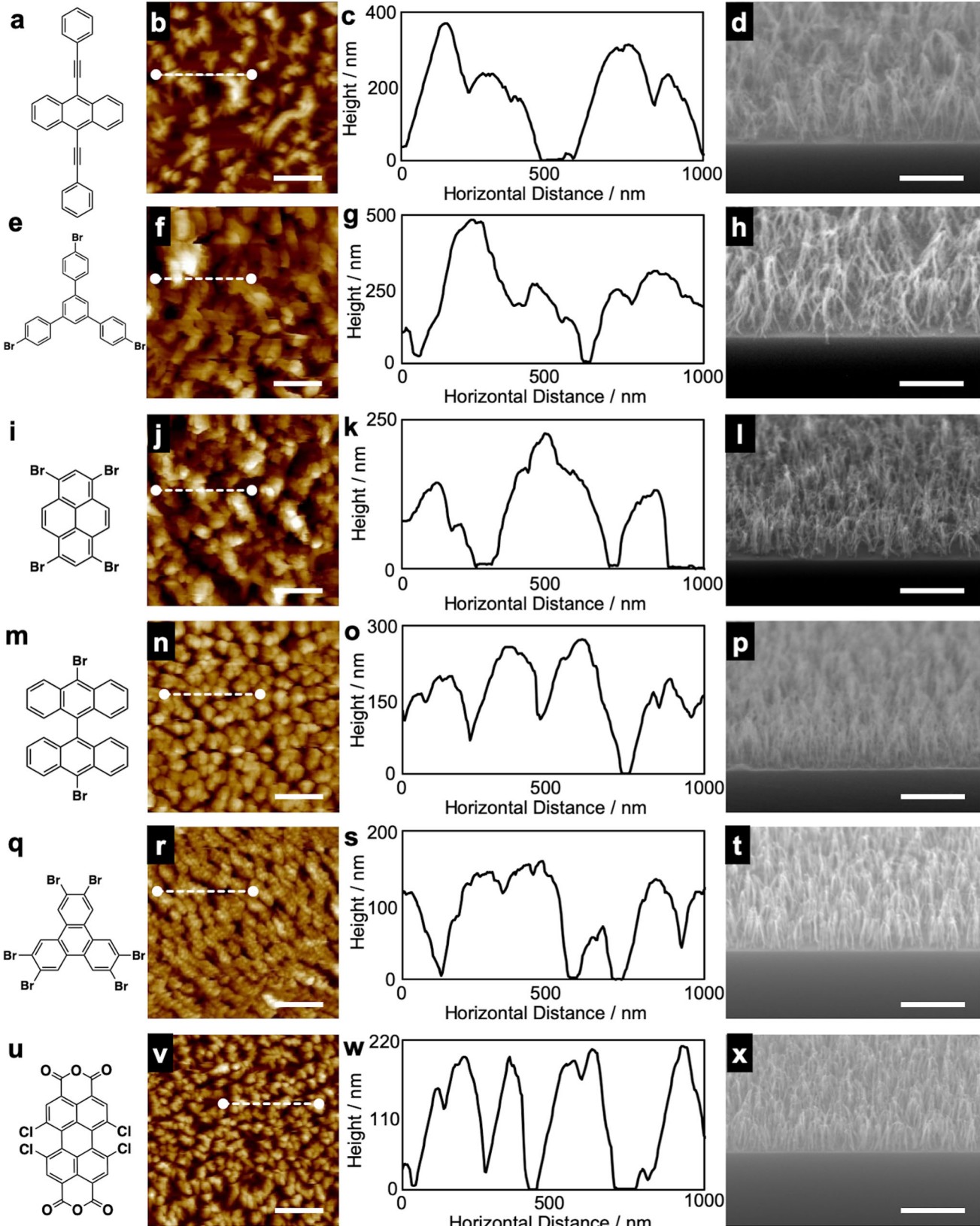

**Fig. 3 Nanowires from various aromatic compounds isolated by dry process. a**, **e**, **i**, **m**, **q**, **u** Chemical structure, **b**, **f**, **j**, **n**, **r**, **v** AFM topographic images and **c**, **g**, **k**, **o**, **s**, **w** height profiles along white dashed lines, and **d**, **h**, **l**, **p**, **t**, **x** SEM images of nanowires fabricated via irradiation of vapor-deposited films of **a–d** 450 nm 9,10-bis(phenylethynyl)anthracene (BPEA), **e–h** 500 nm 1,3,5-Tris(4-bromophenyl)benzene (TBPB), **i–l** 250 nm 1,3,6,8-tetrabromopyrene (TBP), **m–p** 300 nm 10,10′-dibromo-9,9′-bianthracene (DBBA), **q–t** 200 nm 2,3,6,7,10,11-hexabromotriphenylene (HBT), **u–x** 220 nm 1,6,7,12-tetrachloro-3,4,9,10-perylenetetracarboxylic dianhydride (PTCDA-Cl$_4$). Scale bars represent 500 nm. Irradiation conditions: 450 MeV $^{129}$Xe$^{23+}$ at $1.0 \times 10^{11}$ cm$^{-2}$.

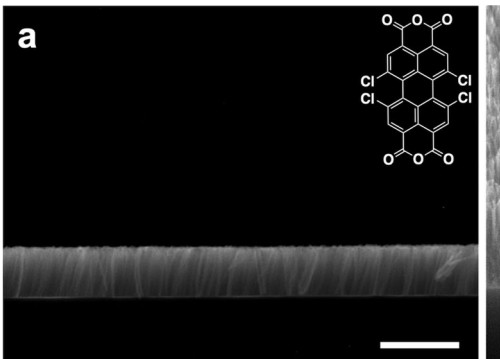
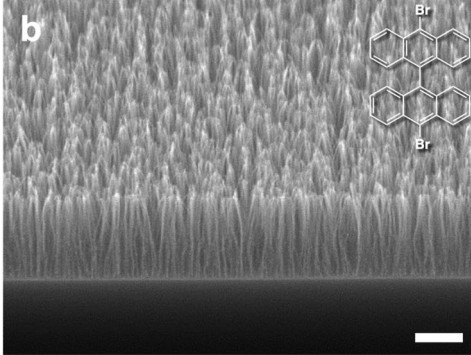

**Fig. 4 Side view of standing nanowires. a** Side view SEM image of free-standing nanowires fabricated via irradiation of 300 nm PTCDA-Cl$_4$ film with 120 MeV $^{197}$Au$^{9+}$ at $1 \times 10^{11}$ cm$^{-2}$ and subsequent sublimation. **b** SEM image of free-standing nanowires fabricated via irradiation of 1300 nm DBBA film with 450 MeV $^{129}$Xe$^{23+}$ at $1 \times 10^{11}$ cm$^{-2}$ and subsequent sublimation. Scale bars represent 500 nm.

bonds[58]. Much higher enhancement of the reactions could be promoted by dissociative electron attachment onto halogen atoms bound to aromatic rings[59,60], resulting in the successful formation of standing nanowires based on all five starting compounds represented in Fig. 3. The cross-sectional radius of the nanowires reflects only the local energy density released by a particle in the STLiP protocol, which is determined physically and given uniformly in principle. In the wet process, the solubility/insolubility of molecules at the nanowire surfaces predominantly determines the radius, and the fluctuations in the dissolution process are reflected in the roughness of the surfaces, and hence the nonuniform values of radii in the nanowire aggregates. In sharp contrast, the standing nanowires based on 1,6,7,12-tetrachloro-3,4,9,10-perylenetetracarboxylic dianhydride (PTCDA-Cl$_4$) isolated by the dry process (Fig. 4a) clearly confirm the high uniformity of their height, which coincides with the initial thickness of the films prepared. A nanowire aspect ratio of over 100 was achieved in 10,10′-dibromo-9,9′-bianthracene (DBBA)-based nanowires isolated by the all-dry process (Fig. 4b). The results suggest the wide versatility of the STLiP method for the conversion of various small organic molecules into free-standing VA-nanowires with high aspect ratio; the starting materials are not limited to C$_{60}$. The important factors of the target molecules for the efficient formation of standing nanowires by this method include (1) aromatic molecules, (2) sublimable at a temperature below the melting point, and (3) with substitution by triple and/or C–X bonds in their periphery.

**Structural parameter analysis of standing nanowires.** SEM/AFM images are beneficial for the characterization of nanowire shapes/morphologies. However, they are insufficient for insights into the internal structures of nanowires, which depend on the target molecules and their polymerization reaction efficiency. We chose the following four parameters: the cross-sectional radius ($r$) and rigidity of each nanowire ($D/L$), the coverage factor per unit area ($C_{NW}$), and the length retention of standing nanowires compared to the initial organic thin film thickness ($H_{NWs}/T_{film}$), for further discussion to shed light on the interior characteristics of nanowires (Supplementary Fig. 6). As represented by Eq. (3), the polymerization reaction efficiency in a particle track explicitly defines the density of polymerization/cross-linking networks in the area, mechanically reinforcing the nanowires. As the flexibility/rigidity of macromolecular chains have been quantitatively discussed[40], the formulations have been scaled herein and their rigidity was determined statistically as $D/L$, where $D$ and $L$ are the end-to-end distance and total length of nanowires, respectively. For all the molecules employed, as summarized in Table 1, the values of $D/L$, which were estimated from AFM images to

knocked-down nanowires at $10^9$–$10^{10}$ cm$^{-2}$, approached ~1, exhibiting the high rigidity of nanowires without a strong dependence on $r$. Since the impact points of particles are uncontrollable at the nm scale and are distributed randomly in the plane of the targets, we can presume the partial overlap of ion tracks under high particle fluence, impacting the uniformity of the nanowires. The Poisson distribution function was used to obtain a numerical estimate for the ion-track overlap[61,62], and the fraction of overlapped area against unoverlapped area is represented by the Poisson measure $C_{NWs}$ as summarized in Table 1. Under the condition of ion irradiation at $1 \times 10^{11}$ cm$^{-2}$, the value of $C_{NWs}$ is ~10%, and no significant overlap occurs, so that uniform nanowire arrays can be assured. SEM and AFM images can precisely estimate the height of nanowires without significant errors. It was observed that the average height of the top of nanowires ($H_{NWs}$) was less than the initial thickness of the target films ($T_{film}$). This observation indicates that the nanowires partially lean to form a nanowire carpet with inter-wire interactions. It should be noted that changes in $H_{NWs}/T_{film}$ show no correlation with the rigidity parameter $D/L$, suggesting that high enough mechanical rigidity had been already secured by chemical reaction prior to the thermal treatments to isolate the nanowires. Therefore, a possible scenario is that the reaction efficiency, $G$ value, and the size of the monomer molecule mainly determine the nanowire radius ($r$) (and $C_{NWs}$), resulting in closer distances between the surfaces of adjacent nanowires. The larger the nanowire radius, the more the nanowires remain vertical with inter-wire interactions.

**Fabrication of nanowire networks.** Directional alignment of nanowire growth relative to a 2D plane has been a challenging task for 1D nanostructured systems, for both bottom-up methods, such as CNT growth in catalytic cycles and top-down methods of nanolithography. Although there have been only limited examples of 3D structure of metal wires at nano (~submicron) scale[63], ultrafine networked structures from organic nanowires have not yet been reported. In this context, the present STLiP method is promising, because the momentum of a charged particle is unchanged in its interactions with the organic matter and is confined to the growth direction of the corresponding nanowires. Thus, once irradiation is performed with a certain tilt angle ($\theta$) against the flat surface of a substrate, tilted nanowires are produced as shown in Supplementary Fig. 7. Upon continuous rotation of a substrate at a fixed $\theta = 45°$, a conical distribution of the tilted nanowires was isolated, showing cross-connected structures with cross-connected scaffolds (Supplementary Fig. 8). Spatial control of the target against the unidirectional momentum of the high-energy charged particles enables us to

**Table 1 Comparison of various parameters of nanowires obtained from sublimable organic small-molecule materials.**

| Materials | $r^a$ (nm) | $C_{NWs}{}^b$ (%) | $D/L^c$ | $H_{NWs}/T_{film}{}^d$ | $G$ value$^e$ (100 eV)$^{-1}$ |
|---|---|---|---|---|---|
| $C_{60}$ | 6.6 ± 0.6 | 12.8 | 0.97 ± 0.03 | 0.97 ± 0.03 | 35 |
| BPEA | 5.0 ± 0.4 | 7.6 | 0.95 ± 0.09 | 0.59 ± 0.11 | 37 |
| TBPB | 4.9 ± 0.4 | 7.3 | 0.94 ± 0.03 | 0.64 ± 0.14 | 31 |
| TBP | 5.3 ± 0.5 | 8.4 | 0.94 ± 0.06 | 0.76 ± 0.13 | 41 |
| DBBA | 6.1 ± 0.6 | 11.0 | 0.94 ± 0.06 | 0.84 ± 0.07 | 47 |
| HBT | 6.1 ± 0.4 | 11.0 | 0.90 ± 0.10 | 0.92 ± 0.07 | 42 |
| PTCDA-Cl$_4$ | 6.1 ± 0.3 | 11.0 | 0.95 ± 0.04 | 0.94 ± 0.04 | 43 |

Irradiation conditions: 450 MeV $^{129}$Xe$^{23+}$ at $1.0 \times 10^{11}$ cm$^{-2}$.
$^a$Average diameter of nanowires ($r$).
$^b$Coverage factor estimated from the ratio of cross-sectional area of nanowires to substrate surface area ($C_{NWs}$).
$^c$Average end-to-end distance of nanowires ($D$) and average length of nanowires ($L$).
$^d$Average height of nanowires ($H_{NWs}$) and initial thickness of films ($T_{film}$).
$^e$Reaction efficiency per 100 eV.

form 3D nanowire networks with designed structures as demonstrated in these figures, suggesting its potential for nano-design with a three-dimensional molecular canvas.

**Formation of heterojunctions in standing organic nanowires.** Surface modification of nanowires[64,65] is important for the functionalization of nanowires toward developing miniaturized devices or biocompatible nanosystems. We consider that the present STLiP method allows the design of any heterointerface with free choice of the two materials by simply employing bilayer structures as the targets. In addition, coaxial heterointerfaces can be attainable by subsequent chemical/physical decoration of the active surfaces of standing-isolated nanowires. Here, we focused on the feasibility of using our standing nanowire arrays to form controlled heterojunctions between $p$- and $n$-types semiconductor organic molecules. We have addressed this issue via the STLiP protocol to form uniform nanowires with programmed heterojunctions in a single step over an arbitrary area.

Titanyl phthalocyanine (TiOPc) and $C_{60}$, as representative $p$- and $n$-type organic molecules[66], respectively, were used to prepare a bilayer thin film. Nanowires with single heterojunctions were prepared by the usual STLiP protocol, controlling the fluence of irradiation at $1 \times 10^{11}$ cm$^{-2}$ (Fig. 5a). The cross-sectional image of the bilayer film exhibited a clear contrast reflecting the composition of the respective molecules with metal and carbon atoms, showing a clear boundary (Fig. 5b). After isolation by sublimation, the nanowires exhibited the expected self-standing features, as confirmed by the SEM images (Fig. 5c, d), and the lengths of the nanowire segments were uniform and consistent with the film thicknesses. The chemical structure of each segment after isolation of the sublimation protocol, and then in free-standing nanowire form, was investigated using Raman spectroscopy (Supplementary Fig. 9), showing the clear signatures of the $C_{60}$ cages and Pc rings. This suggests that these structures were preserved in the nanowires, and hence one heterojunction per nanowire was produced with less damage in this dry process. The choices of two distinct molecular systems enables the free design of heterojunctions between the molecular systems. Another example using PTCDA-Cl$_4$ and HBT was also demonstrated as shown in Fig. 5e–g. Through stepwise sublimation by fine control of the sublimation temperature and degree of vacuum, the isolation process of $p/n$ heterojunction block-co-nanowires was clearly demonstrated. Further accumulation of heterojunction structures in an isolated nanowire was demonstrated under consistent sublimation conditions of $p$ and $n$ components. Multi-segment nanowire arrays have been clearly visualized in Supplementary Fig. 10 from a simple structure of

layer-by-layer films of $C_{60}$ and copper phthalocyanine (CuPc) as a target. This allows us to program a variety of heterojunction structures into a single wire with ordered alignment from favorable choices of sublimable molecular systems.

Standing nanowires with maximized active surfaces allow us to introduce heterojunction structures not only along the longitudinal, but also the radial direction. Uniform overcoating/decoration of 1D nanowires is one of the simplest way to achieve coaxial nanostructures[32–35]. We employed electropolymerization of π-conjugated monomers grown from the semiconducting surfaces of the standing nanowires. Polythiophene, a well-known $p$-type organic semiconductor material[67], can be easily deposited by electropolymerization of the thiophene monomer or dimer[68]. After forming $C_{60}$ nanowires on a conductive ITO-glass substrate, electropolymerization was carried out (Fig. 6a), and a thin film of polythiophene was prepared on the substrate (Fig. 6b). AFM and SEM confirmed that the polythiophene grew from the wire periphery rather than from the ITO substrate, resulting in the formation of nanowires with a coaxial structure (Fig. 6c–h and Supplementary Fig. 11a–c). As seen from the side view of SEM images before/after polymerization (Fig. 6c, d, f, g) the lengths of the nanowires are almost identical, while their diameters got increased apparently. In addition, as electropolymerization progressed, polythiophene was further grown around the $C_{60}$ nanowires in their radial direction (Supplementary Fig. 11d, e). Figures 5 and 6 together show that the wire can be readily extended in the longitudinal and radial directions in a very simple manner, constructing single or coaxial heterointerfaces. Considering that organic heterojunction structures at the nanoscale have been entirely addressed by the epitaxial growth[34,36] or template method[35,38] of one component from the active surface of the other component or bulk heterojunction-type simple mixtures[69] with limited choice of materials, our unique method using standing nanowire platforms serves as a promising strategy for accessing future nanoscale heterojunction devices.

**Discussion**
We report the facile and efficient technique to form standing-isolated organic nanowire arrays with small radii (<10 nm) and vast total surface area, which simultaneously show wide versatility in the choice of organic starting materials and free binary choices for the presumed heterointerfaces, paving the way toward the development of functional nanomaterials for optical, electrical, and biological applications. High-energy particle irradiation of organic thin films in the orthogonal configuration resulted in polymerization/cross-linking reactions of the small organic molecules within the cylindrical ion tracks to give uniform

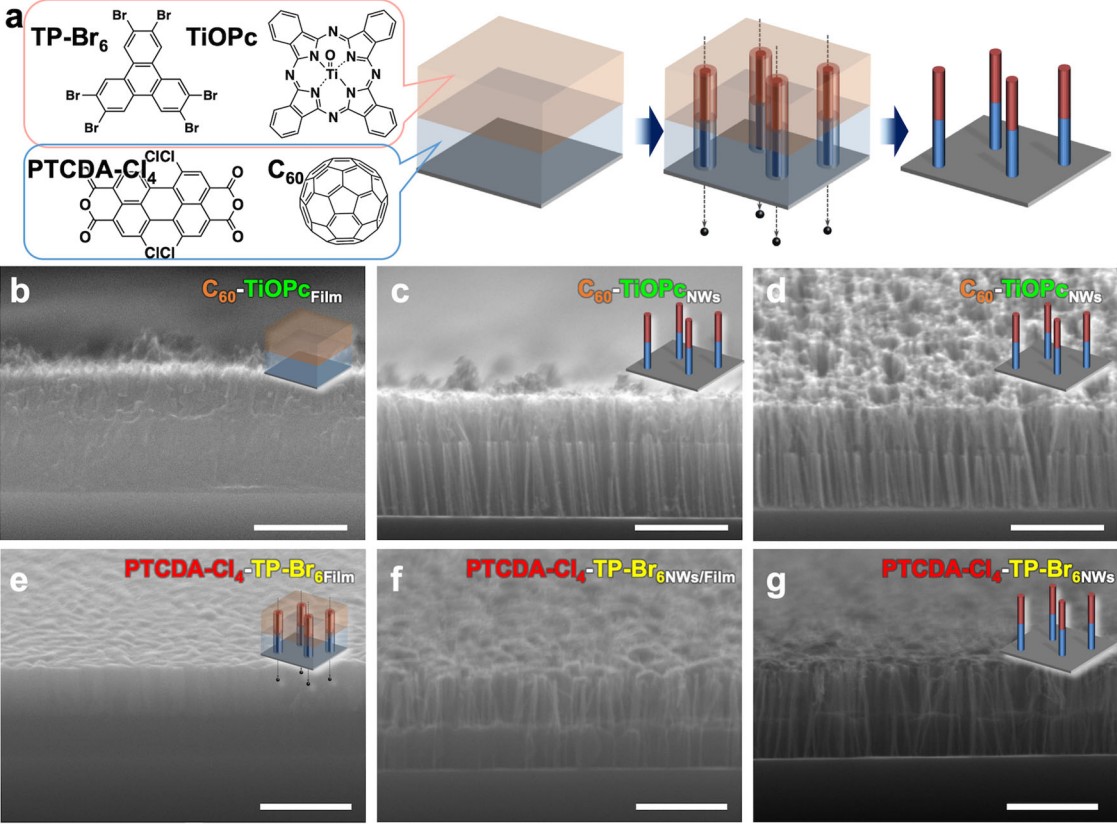

**Fig. 5 Block-*co*-nanowire-type heterointerfaces. a** Schematic illustration of fabrication method of free-standing two-segment nanowires from bilayer films. **b** Side view SEM image of bilayer film of titanyl phthalocyanine (TiOPc) (top) and $C_{60}$ (bottom). **c** Side and **d** tilt views of two-segment nanowires fabricated via irradiation of TiOPc/$C_{60}$ bilayer film with 120 MeV $^{197}Au^{9+}$ at $1 \times 10^{11}$ cm$^{-2}$ and subsequent sublimation at ~300 °C. **e** A SEM image of bilayer film of TP-Br$_6$ (HBT, top) and PTCDA-Cl$_4$ (bottom). **f, g** Tilt views of two-segment nanowires fabricated via irradiation of TP-Br$_6$/PTCDA-Cl$_4$ bilayer film with 120 MeV $^{197}Au^{9+}$ at $1 \times 10^{11}$ cm$^{-2}$ and stepwise sublimation at 187 °C for 5 min and 300 °C for 5 min. Scale bars represent 500 nm.

polymerized nanowires with a defined length and diameter. The key to the achievement is the use of a dry process, sublimation, after irradiation, allowing isolation of the standing organic nanowires with remarkable vertical alignment. The well-aligned nanowires on the substrate exhibited unique functionalities indicating that the structural uniform length and vertical alignment of the free-standing nanowires was advantageous for surface modification, leading to high-water repellency (Supplementary Fig. 12). The well-aligned nanowires on a substrate exhibited unique functionalities reflecting (1) structural: uniform length and vertical alignment of free-standing nanowires was advantageous for surface modification leading high-water repellency (Supplementary Fig. 12)[70], (2) electrical conductivity of the nanowire plexus: the nanowire of conjugated organic molecules was expected to be an electrical conductive wire, and the conductivity of the nanowire plexus was measured by DC conductivity measurement because the vertical alignment allow us to fabricate a sandwiched cell between metal electrodes. The observed I–V traces were shown in Supplementary Fig. 13, suggesting almost linear dependence of conductivity on the number density of nanowires. It should be also noted that the conductivity is also proportional to the area of top electrodes fabricated on the nanowire plexus, suggesting high uniformity of the nanowires also in their electrical conduction, (3) A p–n heterojunction nanowire as a rectifier diode: the heterojunction structure built in the nanowire was presumed to rectify electrical current in the nanowire. The I–V characteristics of the nanowire were traced under an applied modulated bias (Supplementary Fig. 14), suggesting statistically unipolar electrical

conduction over the heterojunction and its potential as an ultrasmall rectifier diode.

This methodology is applicable to a variety of aromatic organic molecules, especially aromatic molecules with triple carbon–carbon or aryl-halogen bonds capable of highly efficient radical generation upon high-energy particle irradiation. Simple irradiation of multilayer organic films provides the fabrication of segmented nanowires with desired heterointerfaces, which enables nanoscale p–n heterojunction nanowires. Electropolymerization of another π-conjugated monomer on the standing nanowire arrays gives coaxially grown nanowires with a heterointerface, while keeping their VA free-standing characteristics. Our simple method of producing standing nanowires with uniform ultrafine small radii and extra-large specific surface area, with free choice of organic molecule starting materials and controllable length and number density, provides an opportunity to construct organic nanowire-based devices contributing to the fields of nano photonics, electronics, mechanics, and sensing with highly efficient and/or anisotropic functions.

## Methods

**Materials**. $C_{60}$, $C_{70}$, PC$_{61}$BM, BPEA, TBPB, TBP, DBBA, HBT, TiOPc, CuPc, and 2,2'-bithiophene were purchased from Tokyo Chemical Industry Co. Ltd. and used without further purification. PTCDA-Cl$_4$ was purchased from Combi Blocks Inc.

**Film preparation**. A Si substrate was cut into 1.5 cm$^2$ squares, sonicated in 2-propanol, dried, and treated with UV-O$_3$ prior to the use. Thin films of PC$_{61}$BM were prepared by spin-coating from CHCl$_3$ solutions (5–10 wt%) on the Si substrate. Thin films of $C_{60}$, $C_{70}$, BPEA, TBPB, TBP, DBBA, HBT, PTCDA-Cl$_4$, TiOPc, and CuPc were prepared by vapor deposition under ~10$^{-4}$ Pa at the rate

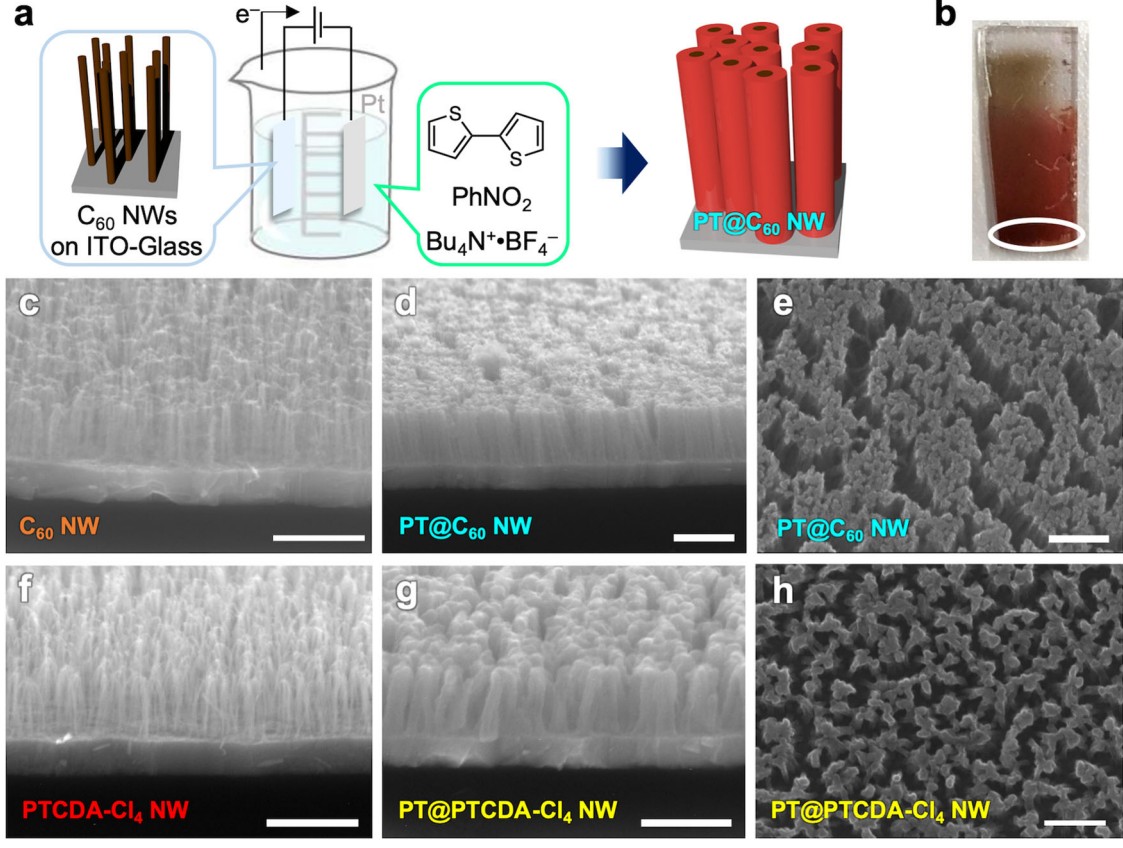

**Fig. 6 Coaxial nanowire-type heterointerfaces. a** Schematic illustration of fabrication protocol for free-standing coaxial nanowires by electropolymerization of bithiophene from $C_{60}$ nanowires on ITO-glass. **b** Example of coaxial nanowire film. SEM images of **c** $C_{60}$-based VA-NWs and **d**, **e** polythiophene-wrapped $C_{60}$-based coaxial nanowires (PT@$C_{60}$) in white circled area in **b**. SEM images of **f** free-standing PTCDA-Cl$_4$-based nanowires and **g**, **h** similarly fabricated PT@PTCDA-Cl$_4$ nanowires. Scale bars represent 500 nm. Irradiation conditions: 120 MeV $^{197}$Au$^{9+}$ at $1.0 \times 10^{11}$ cm$^{-2}$ for $C_{60}$ nanowires; 450 MeV $^{129}$Xe$^{23+}$ at $1.0 \times 10^{11}$ cm$^{-2}$ for PTCDA-Cl$_4$ nanowires.

of 0.2–0.8 Å s$^{-1}$. The thickness of the films was evaluated by a Veeco Instruments Inc. model Dektak 150 surface profiler.

**Irradiation and isolation**. The 490 MeV $^{192}$Os$^{30+}$, 350 MeV $^{129}$Xe$^{26+}$, and 450 MeV $^{129}$Xe$^{23+}$ were generated from a cyclotron accelerator at Takasaki Advanced Radiation Research Institute, National Institutes for Quantum and Radiological Science and Technology. The 120 MeV $^{197}$Au$^{9+}$ ions were generated from a Pelletron accelerator at Inter-University Accelerator Centre. All the charged particles (ions) employed in this study have a higher velocity than the Bohr velocity of electrons in the target materials, and thus the energy of the charged particles is presumed to be transferred by inelastic collision events without specific dependence on the atomic numbers of the particles. The prepared organic films were exposed to the ion beam in a vacuum chamber ($<1 \times 10^{-4}$ Pa). The number of incident particles was controlled at the fluence of $10^9$–$10^{12}$ cm$^{-2}$ by setting the exposure time (s) and flux (cm$^{-2}$ s$^{-1}$) calculated from the beam current, charges of the ions used, and elementary charge. Then, the irradiated films were further cut into small pieces, and developed by immersing them into organic solvents or sublimation under vacuum ($5$–$8 \times 10^{-3}$ Pa). The end of sublimation was judged visually by observing the change in the reflectance of samples.

**Simulation of high-energy particle irradiation**. The loss of the kinetic energy of the ions due to their traversal through the organic films (LET) was estimated using the SRIM 2010 simulation code. The film thickness was set at 10 μm (larger than the actual maximum film thickness). The density of the film was set at 1.72 g cm$^{-1}$ for $C_{60}$ and 1.96 g cm$^{-1}$ PTCDA-Cl$_4$. The 490 MeV $^{192}$Os$^{30+}$, 350 MeV $^{129}$Xe$^{26+}$, 450 MeV $^{129}$Xe$^{23+}$, and 120 MeV $^{197}$Au$^{9+}$ ions were irradiated 2000 times each.

**Morphological characterization of nanowires**. The sizes and shapes of the isolated nanowires were observed using a Bruker Co. model Multimode 8 AFM and JEOL Ltd. JSM-7001F SEM system.

**Characterization of nanowire parameters**. The end-to-end distance ($D$) and length ($L$) of the nanowires were estimated by AFM using the knocked-down

nanowires after irradiation with 450 MeV $^{129}$Xe$^{23+}$ at $1.0 \times 10^9$ cm$^{-2}$ and the subsequent sublimation process. The cross-sectional radius was calculated from the half-width ($r_w$) and half-height ($r_h$) of the AFM images of the knocked-down nanowires. By applying the ellipsoidal model, the cross-sectional radius ($r$) was defined as $r = (r_w r_h)^{1/2}$.

**Raman spectroscopy**. Raman spectra were measured on a JASCO NRS-4100 spectrometer.

**Electropolymerization**. Standing nanowires were fabricated on an ITO-glass substrate, and the NW-coated ITO-glass was used as an anode. The anode and cathode (Pt plate) were dipped in a nitrobenzene solution (30 mL) of 2,2′-bithiophene (131 mg) and tetrabutylammonium tetrafluoroborate (200 mg). A constant electric current of 0.5 mA was applied for 1 min by a Hokuto Denko Corporation HA-151B galvanostat. After the polymerization of 2,2′-bithiophene on the anode, de-doping was carried out by applying the inverse current of 0.5 mA for 1 min. The anode was pulled out from the solution and gently washed by methanol.

**DC conductivity measurements**. The electrical conductivity measurements for $C_{60}$ nanowires in vertical direction were conducted with two-probe configuration. A gold bottom electrode (50 nm) was deposited on a Si substrate by thermal evaporation. Some drops of silver paste were casted onto the $C_{60}$ nanowires leading to top electrodes with various active area. $I$–$V$ traces were recorded using a micromanipulated probe station (Lake Shore TTPX) and a semiconductor parameter analyzer (Keithley 4200A-SCS). All the measurements were performed in vacuum ($\sim 10^{-3}$ Pa) at room temperature.

**Conductive AFM**. Conductivity of PT@$C_{60}$ nanowires were measured by a SII Inc. NanoNavi-II AFM in contact mode with an Au-coated cantilever. Spring constant of the cantilever was 0.23 N m$^{-1}$, and the deflection of the cantilever was set at 2 nm, acting interfacial force of ~400 pN between the cantilever and PT surfaces.

**Evaluation of surface water repellency**. Contact angle measurements were performed on a Kyowa Interface Science Co., Ltd. DMe-211 contact angle meter. A distilled water droplet of 1 μL was dropped on the NW-coated silicon substrate.

## Data availability

The authors declare that the data supporting the findings of this study are available within the paper and its Supplementary Information file. All other information is available from the corresponding authors upon reasonable request.

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

## Acknowledgements

This work was supported in part by a Grant-in-Aid for Transformative Research Areas (20H05862) from Ministry of Education, Culture, Sports, Science, and Technology (MEXT), Japan, Scientific Research (A) (18H03918) and (B) (19KK0134) from the Japan Society for the Promotion of Science (JSPS). High-energy charged particle irradiation was carried out at the Takasaki Advanced Radiation Research Institute and the National Institutes for Quantum and Radiological Science and Technology (QST), Japan, under the Facility Sharing Program and the Inter-University Program for the Joint Use of JAEA/QST Facilities, and at the Inter-University Accelerator Centre (IUAC), Delhi, India.

## Author contributions

S.S. conceived the project with T.S.; K.S.K., K.T.K., T.S., and S.S. co-designed the experiments. K.S.K. and K.T.K. performed nanowire fabrication and characterization experiments. A.I., H.K., M.S., L.G.B.V.S., and D.K.A. co-performed the irradiation experiments; S.G.S. co-analyzed experimental data; M.N. and Y.T. conducted conductive AFM measurements and analysis; M.K. and M.S. examined electrical conductivity of nanowires; and K.S.K., S.G.S., T.S., and S.S. co-wrote the paper.

## Competing interests

The authors declare no competing interests.
