## [Peer Review File · Nature Communications]

REVIEWER COMMENTS

Reviewer #1 (Remarks to the Author):

This manuscript describes the use of high-energy ion irradiation to cross-link films of a variety of aromatic small molecules and synthesize random arrays of vertically-aligned bundled organic nanowires. The novelty of the work comes from the use of sublimation to avoid nanowire collapse at the substrate surface while allowing for the fabrication of high aspect-ratio organic and potentially semiconducting nanowires with sub-10 nm diameters. Achieving this with such small diameters and with the type of organic precursors used in this manuscript is significant. Additionally, the authors report a simple way of preparing nanowires with axial junctions, which should be appealing to a broad audience due to the potential generality of the approach. The manuscript is well-written and the morphology of the materials produced well-characterized via AFM and SEM. However, some of the claims are not adequately substantiated, and some of the materials should be better characterized. Thus, I would recommend publication once the following points have been addressed:

Major comments:

- The authors need to demonstrate the functionality of the axial PN nanowire junctions (block co-nanowires Figure 5). Without functionality, the work presented here is of lesser importance.
- Line 318-319: The authors write: "it was confirmed that one heterojunction per nanowire was produced with no damage in this dry process". The authors should provide both structural and functional evidence for this.
- Lines 323-326: The authors write: "Nanowire arrays with different molecules at the single-molecule level [...] are available by our method". This sentence should be modified. It is unclear and potentially misleading: the authors have provided no evidence that they achieved a synthetic precision at the single-molecule level.
- The authors should investigate and write how thin the nanowire segments can be within the axial heterojunction nanowires (block co-nanowires, Figure 5). Have the authors tried to prepare nanowires with more than 2 segments (i.e. instead of a AB nanowire, prepare a ABABABABA... nanowire)? This would really demonstrate the versatility of the technique.
- Lines 29-30: the authors write that they report the synthesis of nanowires with single-nm thickness. This is incorrect and misleading since the nanowire diameters are > 5 nm.

Minor comments:

- The authors claim that they form radial polythiophene-C60 nanowires as confirmed by AFM and SEM (line 335). The data provided (Figure 6d) is not entirely convincing.
- The authors use different types of ions for the cross-linking. Could they explain the rationale behind it?
- Lines 23-25 (Abstract). The authors write that integrating nanostructures with aspect ratio of 10 is difficult in 3D. I would argue that arrays of nanowires with aspect ratios > 10 can be routinely prepared (silicon nanowires, for example), especially when using supercritical CO₂ drying.
- Figure 5: the legend is incomplete (e.g images are not described)
- The authors should discuss the use of templated synthesis within porous alumina membranes to prepare nanowires with axial organic/organic and radial organic/inorganic junctions.

Comment on the reproducibility of the work: The experiments are adequately described

Gilles R. Bourret

Reviewer #2 (Remarks to the Author):

This study reports the fabrication of vertically aligned organic nanowires by the controlled deposition of energy from heavy ion implantation resulting in linear polymerization of the organic materials present. The resultant nanowires have aspect ratios in excess of 100 and the methodology has allowed the fabrication of heterointerface nanostructures. The prime novelty of the work is associated with removal of unpolymerized phase, which instead of wet chemistry which tends to destroy the vertical alignment, sublimation of the organic phase is used. Although that

might limit the polymer materials that can be used, the approach does produce material that could be important for anisotropic applications e.g. organic electronics, liquid crystals, polymer-based metamaterials. The experimental approach and conclusions drawn appear reasonable and will be of interest in the wider materials physics and chemistry communities as the values of aspect ratio and degree of alignment are impressive and the method itself could be transformational for the production of highly aligned organic materials. I recommend publication following consideration by the authors of the comments below.

1. Readers of the article may appreciate further discussion of whether the approach can independently control the lateral spacing between the nanowires and independently control the diameter. Independent control is important; please re-read the discussion of the coverage factor.
2. Table 1 suggests that the NW radius is 4-6 nm for all the organics explored; can this be tailored to larger values?
3. On line 42 there is reference to maximizing the in-plane density; it is important to note that if the goal is to maximise electrical conductivity this could be achieved by having fewer but wider diameter conductors. As such I would encourage the authors to think about this wider point.
4. Finally some comment on the wider applicability of the approach should be included; 450 MeV Os ion implantation is not that common!

~ Point-by-point response to the reviewer comments ~

Reviewer #1 (Remarks to the Author):

This manuscript describes the use of high-energy ion irradiation to cross-link films of a variety of aromatic small molecules and synthesize random arrays of vertically-aligned bundled organic nanowires. The novelty of the work comes from the use of sublimation to avoid nanowire collapse at the substrate surface while allowing for the fabrication of high aspect-ratio organic and potentially semiconducting nanowires with sub-10 nm diameters. Achieving this with such small diameters and with the type of organic precursors used in this manuscript is significant. Additionally, the authors report a simple way of preparing nanowires with axial junctions, which should be appealing to a broad audience due to the potential generality of the approach.

The manuscript is well-written and the morphology of the materials produced well-characterized via AFM and SEM. However, some of the claims are not adequately substantiated, and some of the materials should be better characterized. Thus, I would recommend publication once the following points have been addressed:

[Answer] All the comments, in principle, on the functionalities of the current vertically aligned nanowires are of particular importance. We have addressed all the points raised by this reviewer within the document, and hope the additional experimental data and discussions given therein as satisfactorily answers the queries.

Major comments:

- The authors need to demonstrate the functionality of the axial PN nanowire junctions (block co-nanowires Figure 5). Without functionality, the work presented here is of lesser importance.

[Answer] We agree that the functionality of the PN nanowire will be key for the further development of present nanowire fabrication techniques. Other than vertical alignment, the key features of the present technique, are uniformity in the sizes and shapes of the nanowires and the free combination of materials. Of course, the latter is used to fabricate PN junction structures. In addition to the latter, we have demonstrated the functionality of the present nanowire plexus with additional experiments, as mentioned in this response letter.

1) **Structural functionality (vertical alignment and uniformity)**

The uniform length with a free-standing nanowire plexus provides a flat plane for the nanowire heads. This allows for the use of the air-rich “top” layer surface to control surface tension. After fabricating the condensed nanowire plexus on the substrate, we measured the water droplet contact angle on the substrate before and after nanowire fabrication.

Figure. Difference of surface water repellency controlled by orientation of nanowires. AFM top-view images of BPEA nanowires (a) knocked-down on a substrate after development with *n*-decane and (c) standing on a substrate isolated by sublimation. SEM top-view images of C₆₀ nanowires (e) knocked-down on a substrate after development with 1,2-dichlorobenzene and (g) standing on a substrate isolated by sublimation. (b,d,f,h) Sectional images of water droplets on nanowire surfaces corresponding to images (a,c,e,g), respectively. The nanowires were fabricated via irradiation with 120 MeV ¹⁹⁷Au⁹⁺ at 10¹¹ cm⁻².

Among the sublimable molecules examined in this work, we chose BPEA as the target because of the hydrophobic nature of the molecule. For example, as seen in the figure, the knocked-down BPEA nanowires on the substrate showed moderate water-repelling properties (contact angle: $83.7^\circ \pm 1.9^\circ$), and a significantly larger contact angle was observed for the free-standing nanowires ($127.1^\circ \pm 1.3^\circ$). It is worth noting that the number density and composition of the nanowires were identical. Thus, a considerably high water repellency was achieved for the free-standing nanowires (the lotus effect, for example, *Chem. Rev.* **2015**, *115*, 8230–8293), possibly because of the air layer present among the upright nanowires. These results have been added to the Supporting Information as **Fig. S12**, and the related discussions were added to the main text (in page 15) as follows:

“The well-aligned nanowires on the substrate exhibited unique functionalities indicating that the structural uniform length and vertical alignment of the free-standing nanowires was advantageous for surface modification, leading to high-water repellency (Fig. S12).”

2) Electrical functionality (vertical alignment and uniformity)

The vertically aligned uniform length allows for the facile measurement of the electrical conductivity of the nanowire plexus, and allows for the qualitative analysis of the average conductivity of a single wire based on the uniform thickness. The following figure represents the *I-V* trace of the C₆₀ and HBT nanowires fabricated on the Au/Si substrate with a number density of 10×10^{10} and 1.0×10^{12} cm⁻², respectively, suggesting a clear increase in electrical conductivity with an increase in the number of nanowires.

Figure. Electrical conductivity of nanowires. (a) I - V traces of C_{60} nanowire plexus recorded for a variety of number density of nanowires from 1×10^{10} to $1 \times 10^{12} \text{ cm}^{-2}$. The enlarged traces for a non-irradiated control thin film and nanowires with the lower number density were displayed in the superimposed one. (b) Dependence of electrical current under 1V bias applied on the area of top electrodes for the nanowire plexus with the number density of (blue) 1×10^{10} and (red) $1 \times 10^{12} \text{ cm}^{-2}$. Superimposed figure is a logarithmic plot of the current against area of the electrodes.

The results have been incorporated to the Supporting Information as Fig. S13, and the related discussions were added to the main text (in page 15) as follows:

“Electrical conductivity of the nanowire plexus: The nanowire of conjugated organic molecules was expected to be an electrical conductive “wire”, and the conductivity of the nanowire plexus was measured by DC conductivity measurement because the vertical alignment allow us to fabricate a sandwiched cell between metal electrodes. The observed I - V traces were shown in Fig. S13, suggesting almost linear dependence of conductivity on the number density of nanowires. It should be also noted that the conductivity is also proportional to the area of top electrodes fabricated on the nanowire plexus, suggesting high uniformity of the nanowires also in their electrical conduction.”

The results suggest that the nanowires produced in the study are clearly electrically conductive, and based on these results and as suggested by the reviewer, we have started working on our next study, namely, the feasibility study of nano p - n heterojunction structures for electrical current control.

3) Electro-optical functionality of p - n junctions

The electrical conductivity of single-segment nanowires requires the application of a nanowire with a heterojunction as an element of an electrical circuit. By mimicking the device structure of an electrical diode, the unipolar electrical conductivity of PT@C_{60} nanowires can be traced. The observed I - V traces are shown in the following figure (c):

Figure. p - n heterojunction nanowires as a rectifier diode. (a) Schematic illustration of conductivity measurement of a PT@C₆₀ nanowire by contact-mode AFM with an Au coated cantilever. Spring constant of the cantilever was 0.23 N m⁻¹, deflection of the cantilever was set at 2 nm, acting interfacial force of ~400 pN between the cantilever and PT surfaces. (b) Top view of PT@C₆₀ nanowires. (c) I - V traces of PT@C₆₀ nanowires measured at 30 distinctive positions in (b).

As visualised in the above I - V traces, the electrical current depended on the polarity of the applied bias which showed unsymmetrical curves. A feasible current rectifying behaviour was observed at certain points in the PT@C₆₀ nanowire arrays, which depended on the thickness of the PT layers overcoated onto the C₆₀ nanowire cores. A larger electrical current was observed under the applied positive bias, which is consistent with the p - n sequential structures of C₆₀ and PT. The results of a p - n heterojunction nanowire as a 1D “diode” are included in the Supporting Information as **Fig. S14**, and the related discussions were added to the main text (in page 15) as follows:

“A p - n heterojunction nanowire as a rectifier diode: the heterojunction structure built in the nanowire was presumed to rectify electrical current in the nanowire. The I - V characteristics of the nanowire were traced under an applied modulated bias (Fig. S13), suggesting statistically unipolar electrical conduction over the heterojunction and its potential as an ultrasmall rectifier diode.”

- Line 318-319: The authors write: “it was confirmed that one heterojunction per nanowire was produced with no damage in this dry process”. The authors should provide both structural and functional evidence for this.

[Answer] After the fabrication of nanowires on the substrate, we had access to several available structural analysis methods, however, only Raman spectroscopy had sufficiently high sensitivity. The Raman spectra of the C₆₀ and copper phthalocyanine nanowires were recorded and presented in the document as follows:

Figure. Raman spectra of (a) C_{60} film, (b) CuPc film, and (c) CuPc- C_{60} nanowires on Si substrates. The film of CuPc or C_{60} was prepared by depositing them on Si substrates at 100 nm thick respectively. CuPc- C_{60} nanowires were fabricated by irradiation with 450 MeV $^{129}\text{Xe}^{23+}$ particles at the fluence of $5.0 \times 10^{11} \text{ cm}^{-2}$, followed by dry process development. The characteristic peaks represent (a) A_g mode (1470 cm^{-1}) of C_{60} , and (b) A_{1g} mode (1336 cm^{-1} , 1552 cm^{-1}) of CuPc.

Fortunately, the signatures of the C_{60} cages and Pc ring structures were clearly visible in the spectra of the nanowires, even after isolation by sublimation processes, and this makes the case for suggesting that the molecular structures are preserved even after nanowire fabrication. As the your have suggested, these findings were included in the revised version of the manuscript as **Fig. S9** and is discussed in the main text (on page 13) as follows:

“The chemical structure of each segment after isolation of the sublimation protocol, and then in free-standing nanowire form, was investigated using Raman spectroscopy (Fig. S9), showing the clear signatures of the C_{60} cages and Pc rings. This suggests that these structures were preserved in the nanowires, and hence one heterojunction per nanowire was produced with less damage in this dry process.”

Please note that we have replaced “no damage” with “less damage”.

- Lines 323-326: The authors write: “Nanowire arrays with different molecules at the single-molecule level [...] are available by our method”. This sentence should be modified. It is unclear and potentially misleading: the authors have provided no evidence that they achieved a synthetic precision at the single-molecule level.

[Answer] Thank you for this suggestion. We completely agree with this opinion after reading the initial manuscript again. This sentence has been deleted from the revised manuscript.

- The authors should investigate and write how thin the nanowire segments can be within the axial heterojunction nanowires (block co-nanowires, Figure 5).

[Answer] It is important to quantitatively discuss the efficiency of reactions induced within a charged particle trajectory, and we agree with your suggestion on the importance of the assessment protocols to determine the nanowire thickness. The values were derived in two ways, namely, direct observation by SEM and detailed analysis by AFM traces of the wire

collapsed onto the substrate. The former gave rough estimates of the thickness; however, it was associated with larger errors. Thus, for the quantitative discussions, we used the values from the latter. We have updated the Methods section by adding the process that was used to determine the radius of the nanowires.

“The end-to-end distance (D) and length (L) of the nanowires were estimated by AFM using the knocked-down nanowires after irradiation with 450 MeV $^{129}\text{Xe}^{23+}$ at $1.0 \times 10^9 \text{ cm}^{-2}$ and the subsequent sublimation process. The cross-sectional radius was calculated from the half-width (r_w) and half-height (r_h) of the AFM images of the knocked-down nanowires. By applying the ellipsoidal model, the cross-sectional radius (r) was defined as $r = (r_w r_h)^{1/2}$.”

- Have the authors tried to prepare nanowires with more than 2 segments (i.e. instead of a AB nanowire, prepare a ABABABABA... nanowire)? This would really demonstrate the versatility of the technique.

[Answer] We deeply appreciate the reviewer for this insightful demonstration. As per your suggestion, we prepared multi-layered organic films by the vapour deposition of representative electron-donating and electron-accepting molecules, namely, C_{60} and copper phthalocyanine (CuPc) molecules. The criterion for choosing these molecules is the matching of vapour pressure for sublimation in the present all-dry process. We presumed that the mismatch in vapour pressure might cause the rapid vaporisation of the underlayers in the protocol, leading to the collapse of the nanowires. A small mismatch was still present in the pressure of C_{60} and CuPc. These results were surprising. As you suggested, the multi-layered film yielded clear multi-segment free-standing nanowires, as shown in the figure below, and the small mismatch did not cause any significant damage to the nanowires.

Figure. (a,b) SEM images of multi-segment nanowires fabricated by irradiation of 450 MeV Xe ions to CuPc and C_{60} film with 10 layers (20 nm thick for each layer, total 200 nm thick). Clear bright tone is suggestive of Cu atoms in CuPc layers in contrast to C_{60} layers with only light carbon atoms.

We believe that this shows the high feasibility of the technique for nanowire fabrication. The successful formation of multi-segment nanowires with sizes of a few \sim tens nm in both the axial and radial directions inspired us to use and check the quantum confinement of electrons and holes for further functionality of the present nanowire systems. We thank you for inspiring

us with such an idea for our next study. We hope to report the successful demonstration of nanowires as quantum devices in the near future.

The results of the multi-segment nanowire plexus were included in the Supporting Information as **Figure S10**, and related discussions are given in page 13 as follows:

“Further accumulation of heterojunction structures in an isolated nanowire was demonstrated under consistent sublimation conditions of p and n components. Multi-segment nanowire arrays have been clearly visualized in Fig. S10 from a simple structure of layer-by-layer films of C₆₀ and copper phthalocyanine (CuPc) as a target. This allows us to program a variety of heterojunction structures into a single wire with ordered alignment from favorable choices of sublimable molecular systems.”

- Lines 29-30: the authors write that they report the synthesis of nanowires with single-nm thickness. This is incorrect and misleading since the nanowire diameters are > 5 nm.

[Answer] As the referee highlighted, single-nm thickness is wrong based on Table 1 (single-nm radius is correct). We have corrected the expression as *“approximately 10~15 nm thickness”*.

Minor comments:

- The authors claim that they form radial polythiophene-C₆₀ nanowires as confirmed by AFM and SEM (line 335). The data provided (Figure 6d) is not entirely convincing.

[Answer] **Fig. 6c** and **6d** (**Fig. 6f** and **6g**) are the SEM images of the same sample before and after the electropolymerisation of bithiophene. As seen in these images, the lengths of the nanowires were almost identical, while their diameters increased. The images in **Fig. S11** also show the signs of radial extension by polymerisation. We hope that this evidence satisfies the referees and readers.

- The authors use different types of ions for the cross-linking. Could they explain the rationale behind it?

[Answer]

We acknowledge your reminder for us to detail, for the readers, the intrinsic nature of the charged particle interaction with matter. From the conclusion, it is not important to use the different atomic numbers of the incident particles (ions), if the particles have sufficiently high LET values. We intended to vary the atomic numbers of the particles to prove the concept, thus demonstrating the feasibility of the present technique.

Background: The energy transfer process from high-energy particles injected into solids is roughly divided into two different energy loss pathways, namely, the energy transfer by direct collision of the particle with the target nuclei (nuclear stopping or elastic energy loss) and the energy transfer by glancing collision events (Coulombic interaction) with the target electrons (electronic stopping or inelastic energy loss). The relative contributions of the above two pathways simply depend on the velocity of an incident particle, the charge states of the particle, and the target atom compositions with an average atomic number of Z_1 . At a velocity, v ,

significantly lower than the Bohr velocity of electrons, v_0 , in the target atoms, the particle tends to be neutralised by electron capture from the target, where elastic collisions are dominant with the target nuclei. At $v > v_0 Z_1^{2/3}$, electrons bearing the incident ion are stripped, reaching the equilibrium charged states. This was the case for our irradiation conditions. In this case, electronic stopping becomes dominant, and was subsequently formulated by Bethe and Bloch as

$$S_e = \frac{4\pi Z_1^2 e^4 n_e}{m_e v^2} \ln \left(\frac{2m_e v^2}{I} \right),$$

where n_e , m_e , and I are the electron density, electron mass, and average excitation energy of the target, respectively.

In the STLiP protocol, the key factor for nanofabrication by particle irradiation is the generation of sufficient radical species and the subsequent polymerization/crosslinking reactions within the ion tracks to avoid dissolution or sublimation in the development processes. Given that nuclear stopping is dominant only in a limited area of the sub-nm radial distance and that v is not in the relativistic regime ($<10\%$ of light speed), only electronic stopping needs to be considered for the subsequent chemical reaction efficiency. Based on the Thomas-Fermi picture of the atom, the particle charge fraction or the effective ion charge, Z_{eff} , is given by:

$$\frac{Z_{\text{eff}}}{Z_1} = \frac{v}{v_0 Z_1^{2/3}}.$$

From this equation, all the charged particles employed hereiherein, namely, $-490 \text{ MeV } ^{192}\text{Os}^{30+}$, $350 \text{ MeV } ^{129}\text{Xe}^{26+}$, and $450 \text{ MeV } ^{129}\text{Xe}^{23+}$, have similar effective ion charges, resulting in the mass and/or initial charge states of the particle being trivial in electronic stopping and, hence, in the efficiency of chemical reactions, caused by the particle. Consequently, we focused only on the LET values for the analysis of the nanowires by the STLiP method. We have clarified this in the Methods section as follows:

“All the charged particles (ions) employed in this work have a higher velocity than the Bohr velocity of electrons in the target materials, and thus the energy of the charged particles is presumed to be transferred by inelastic collision events without specific dependence on the atomic number of the particles.”

- Lines 23-25 (Abstract). The authors write that integrating nanostructures with aspect ratio of 10 is difficult in 3D. I would argue that arrays of nanowires with aspect ratios > 10 can be routinely prepared (silicon nanowires, for example), especially when using supercritical CO_2 drying.

[Answer] We agree with your suggestion. We have revised the sentence in the Abstract of the document as follows:

“Extension into 3D space is a promising future strategy, however, the surface interaction in 3D nanospace hinders the integration of nanostructures with ultra-high aspect ratios.”

- Figure 5: the legend is incomplete (e-g images are not described)

[Answer] Your suggestion will comprehensively improve our manuscript. We selected representative references on axial (segmented) organic/organic heterojunctions (Guo, Y., Li, Y., Li, Y., Liu, H., Li, G., Zhao, Y., & Lin, H. Construction of heterojunction nanowires from polythiophene/polypyrrole for applications as efficient switches. *Chem. Asian J.* **6**, 98–102 (2011)) and radial (coaxial) organic/inorganic junctions (Kovtyukhova, N. I., Kelley, B. K., & Mallouk, T. E. Coaxially gated in-wire thin-film transistors made by template assembly. *J. Am. Chem. Soc.* **126**, 12738–12739 (2004)) using AAO templates. Some sentences in the Introduction, page 4, have been revised as follows:

“Two basic types of heterointerface nanostructures have been proposed thus far, namely, coaxial design (32–35) and two adjoining segments (32,33,36–38). The reported studies are categorised into AAO templated (35,37,38) and physical vapour transport (34,36) methods. From another perspective, they are classified into inorganic–inorganic (32), organic–inorganic (35,37), and organic–organic (34,36,38) nanowire systems.”

Reviewer #2 (Remarks to the Author):

This study reports the fabrication of vertically aligned organic nanowires by the controlled deposition of energy from heavy ion implantation resulting in linear polymerization of the organic materials present. The resultant nanowires have aspect ratios in excess of 100 and the methodology has allowed the fabrication of heterointerface nanostructures. The prime novelty of the work is associated with removal of unpolymerized phase, which instead of wet chemistry which tends to destroy the vertical alignment, sublimation of the organic phase is used. Although that might limit the polymer materials that can be used, the approach does produce material that could be important for anisotropic applications e.g. organic electronics, liquid crystals, polymer-based metamaterials. The experimental approach and conclusions drawn appear reasonable and will be interest in the wider materials physics and chemistry communities as the values of aspect ratio and degree of alignment are impressive and the method itself could be transformational for the production of highly aligned organic materials. I recommend publication following consideration by the authors of the comments below.

[Answer] We intended to extend the feasibility of radiation (high-energy charged particles) to induce non-homogeneous chemical reactions to fabricate nanowires from a variety of “ubiquitous” organic molecules. Your valuable comments motivated us to extend the present processes for future nanomaterial fabrication.

1. Readers of the article may appreciate further discussion of whether the approach can independently control the lateral spacing between the nanowires and independently control the diameter. Independent control is important; please re-read the discussion of the coverage factor.

[Answer] In our method, the average lateral spacing among the nanowires was primarily determined by the number density of the nanowires (irradiation fluence, cm^{-2}), while the diameter can be finely tuned by the LET values by selecting the incident particles. The former

Department of Molecular Engineering, Graduate School of Engineering

Kyoto University Katsura Campus, Nishikyo-ku, Kyoto 615-8510, JAPAN
TEL: +81-75-383-2572 FAX: +81-75-383-2572

can be independently controlled by the total fluence of the irradiated particles (the number of hitting points by the particle) and the latter by the LET value of the particle, allowing us to precisely modulate the lateral interspaces of the free-standing nanowires. In particular, the mechanisms of lateral fine-tuning by incident particles (with tuned LET values) are now discussed in the Methods section as follows:

“All the charged particles (ions) employed in this study have a higher velocity than the Bohr velocity of electrons in the target materials, and thus the energy of the charged particles is presumed to be transferred by inelastic collision events without specific dependence on the atomic numbers of the particles.”

2. Table 1 suggests that the NW radius is 4-6 nm for all the organics explored; can this be tailored to larger values?

[Answer] Further to the answer to the previous question, the radius of the nanowires can be tuned by the LET of the irradiating ions, but the maximum radius will be below 10 nm. Alternatively, the lateral extension of nanowires by electropolymerization, as demonstrated in Fig. 6, works well for enlarging the diameter of the nanowires. In principle, the radius is controlled by controlling the electropolymerization duration.

3. On line 42 there is reference to maximizing the in-plane density; it is important to note that if the goal is to maximise electrical conductivity this could be achieved by having fewer but wider diameter conductors. As such I would encourage the authors to think about this wider point.

[Answer] We appreciate your constructive suggestion for our future work on increasing the electrical conductivity of nanowire systems. We have recognised questions 1 and 2 regarding the increase of the nanowire radius by independent parameter control. In the future, we would like to systematically modulate the nanowire diameters by fine-tuning the diameter of the nanowire cores and the electrical polymerisation of the standing nanowires. To demonstrate the electrical conductivity of the present nanowire systems, we attempted to measure the electrical conductivity and rectify the electrical current in the nanowires with heterojunctions:

Figure. *p-n* heterojunction nanowires as a rectifier diode. (a) Schematic illustration of conductivity measurement of a PT@C₆₀ nanowire by contact-mode AFM with an Au coated cantilever. Spring constant of the cantilever was 0.23 N m⁻¹, deflection of the cantilever was

set at 2 nm, acting interfacial force of ~ 400 pN between the cantilever and PT surfaces. (b) Top view of PT@C₆₀ nanowires. (c) I - V traces of PT@C₆₀ nanowires measured at 30 distinctive positions in (b).

Inspired by your suggestions, we plan to construct all the key electrical nanodevices, including capacitors, resistances, diodes, transistors, and others, with the present technique. Thank you for this inspiration.

4. Finally some comment on the wider applicability of the approach should be included; 450 MeV Os ion implantation is not that common!

[Answer] It is true that Os ions are uncommon for ion implantation and ion irradiation; therefore, we have not used Os ions in our experiment for several years. However, our STLiP method has an excellent choice of irradiated ion species which have a sufficiently high LET regardless of its charge or atomic mass, and we have added further content to the Methods section in this regard.

The use of an accelerator setup may not be the general case for the commercial mass production of materials as there may be a barrier to the implementation of accelerator setups in material fabrication. We achieved a high throughput in the STLiP fabrication protocol, that is, one particle yields one nanowire, which is an extremely efficient process. For instance, to provide a nanowire plexus on a substrate with a number density of 10^{10} cm⁻² at any length up to ~ 10 μ m, only a few seconds were needed to form the nanowires over 10 cm² with a 1 μ A charged particle beam. Any accelerator setup can be operated with the beam current. Charged particles with high energy (equivalent to that used in our study) are, however, very common in the mass production of semiconductors to implant ions in the inorganic semiconductor crystal structures, resulting in atomic doping. Such a set of apparatus is, in principle, applicable to the present STLiP protocols with multivalent ions (and hence, high enough kinetic energies). We were surprised that the “ion implantation” equipment costs were lower than the current cutting-edge nanolithography steppers with extreme-UV and/or excimer laser immersion optics. Even MeV-order high-energy accelerator setups are rather “low-cost” as compared to such an extremely costly apparatus. Numerous high energy accelerator setups have become available globally because of their effectiveness/wide feasibility in cancer radiotherapy. Such setups will also be candidates for our STLiP protocols. Our view of the future, through the present study, is to cause a paradigm shift in the design and fabrication of nanomaterials by using “accelerators”.

REVIEWERS' COMMENTS

Reviewer #1 (Remarks to the Author):

The authors have appropriately addressed all of my comments. I recommend publication of the revised manuscript. Just one last suggestion, the caption of the new Fig. S13b should be corrected: The conductance is plotted against the contact area, not the current, as it is currently written in the caption.

Gilles R. Bourret

Reviewer #2 (Remarks to the Author):

The authors have addressed to the concerns of reviewer 2. The subject material is topical and I recommend publication.

~ Point-by-point response to the reviewer comments ~

Reviewer #1 (Remarks to the Author):

The authors have appropriately addressed all of my comments. I recommend publication of the revised manuscript. Just one last suggestion, the caption of the new Fig. S13b should be corrected: The conductance is plotted against the contact area, not the current, as it is currently written in the caption.

Gilles R. Bourret

[Answer]

Dear Prof. Bourret,

Thank you for your insightful comments in the revising process, as well the recommendation in the caption of the new Figure S13b. We have corrected it as:

Fig. S13. Electrical conductivity of nanowires. (a) I - V traces of C_{60} nanowire plexus recorded for a variety of number density of nanowires from 1×10^{10} to 1×10^{12} cm^{-2} . The enlarged traces for a non-irradiated control thin film and nanowires with the lower number density were displayed in the superimposed one. (b) Dependence of differential conductance under 1 V bias applied on the area of top electrodes for the nanowire plexus with the number density of (blue) 1×10^{10} and (red) 1×10^{12} cm^{-2} . Superimposed figure is a logarithmic plot of the conductance against area of the electrodes.

Reviewer #2 (Remarks to the Author):

The authors have addressed to the concerns of reviewer 2. The subject material is topical and I recommend publication.

[Answer] We are pleased that our revisions made in this manuscript are satisfactory for the reviewer.